# Z-Shaped Electrothermal Microgripper Based on Novel Asymmetric Actuator

**DOI:** 10.3390/mi13091460

**Published:** 2022-09-03

**Authors:** Margarita Tecpoyotl-Torres, Pedro Vargas-Chable, Jesus Escobedo-Alatorre, Luis Cisneros-Villalobos, Josahandy Sarabia-Vergara

**Affiliations:** 1Centro de Investigación en Ingeniería y Ciencias Aplicadas (IICBA-CIICAp), Instituto de Investigación en Ciencias Básicas y Aplicadas, Universidad Autónoma del Estado de Morelos, Cuernavaca, Morelos 62209, Mexico; 2Facultad de Ciencias Químicas e Ingeniería (FCQeI), Universidad Autónoma del Estado de Morelos, Cuernavaca, Morelos 62209, Mexico; 3Licenciatura en Tecnología con Área Terminal en Física Aplicada, Universidad Autónoma del Estado de Morelos, Cuernavaca, Morelos 62209, Mexico

**Keywords:** Ansys™, FEM, MEMS, electrothermal actuation, microgripper, microactuator, chevron

## Abstract

Based on a V-shaped microactuator with a pair of beams, modifications were made to the length and width of a microactuator to observe the effects. A theoretical approach and numerical characterization of the modified microactuator were performed. Its performance was compared to a similar microactuator with equal beam widths, and a V-shaped microactuator. The proposed microactuator, fed at 2 V, compared to the V-shaped actuator, showed a 370.48% increase in force, but a 29.8% decrease in displacement. The equivalent von Mises stress level increased (until 74.2 MPa), but was below the silicon ultimate stress. When the modified microactuator was applied to the proposed microgripper, compared to the case using a V-shaped actuator, the displacement between the jaws increased from 0.85 µm to 4.85 µm, the force from 42.11 mN to 73.61 mN, and the natural frequency from 11.36 kHz to 37.99 kHz; although the temperature increased, on average, from 42 °C up to 73 °C, it is not a critical value for many microobjects. The maximum equivalent von Mises stress was equal to 68.65 MPa. Therefore, it has been demonstrated that the new modified microactuator with damping elements is useful for the proposed microgripper of novel geometry, while a reduced area is maintained.

## 1. Introduction

The development of microelectromechanical systems, MEMS, has found a strong position in various fields of knowledge, opening areas of opportunity for the development of new technological applications. This allows for the generation of new or optimized gyroscopes [1], accelerometers [2], microactuators [3], micropositioners [4], micromirrors [5], microswitches [6], microgrippers [7], microcantilevers [8,9], piezoelectric devices [10], magnetic microactuators [11,12], microgenerators [13,14,15], micropumps [16], and RF-MEMS [17,18], among others.

Understanding the mechanical, electrical, thermic, chemical, and electromagnetic properties of materials, which are compatible with MEMS technology, as well as the theoretical foundations of basic and complex devices, allows for the generation or optimization of systems, in addition to the creation of new fields of knowledge and applications.

In this fascinating field of microsystems, microactuators have received special attention due to their fundamental role in systems developed for several applications. Among them, the following devices stand out.

The first device is a vertical thermal actuator, VTA. A classical VTA is presented in [19], in which the hot (thinner) arm sits on top of the cold (wider) arm, separated by a layer of air. The arms are joined at their ends by a “via”, while at the other ends, they are independently anchored to the substrate. The cold arm is connected to the anchors by means of beams that facilitate its movement in the vertical direction. In [20,21], the VTA is fed by a voltage source. As the hot arm expands, at a greater rate than the cold arm, it pushes the actuator tip down towards the substrate, generating orthogonal displacements.

The U-beam can be considered an improved version of the VTA. It has a pair of arms distributed in parallel connection, forming a U-shaped geometry. The thin arm is kept hot, and the thick arm is characterized by being cold when an electric potential is applied, generating a bending or displacement, and a force at the junction tips of these beams [22,23].

The well-known V-shaped microactuator has a beams distribution, with an inclination angle with respect to the anchor and the shaft. The electric potential applied at their anchors generates the electrothermal deformation of the beams, and the linear displacement of the shaft, as well as the reaction force [24,25,26].

The Z-beam actuators have a similar configuration to the V-shaped actuator, with Z-shaped beams instead of uniform ones. With this modification, an increase in displacement has been reported [27]. They are also actuated by temperature gradients [28,29].

In this paper, a novel microactuator is proposed, with a geometry based on a modified V-shaped actuator. It is characterized by its asymmetrical beams, with different lengths and widths, that generate a higher force response. A design trade-off is observed in the decrease in displacement. To validate the performance of the proposed actuator, a novel micro-gripper, with a Z-shaped arm, was developed. The force improvement was the challenge and motivation for the development of this work

In Table 1, the main characteristics studied in the state of the art for V-shaped and Z-beam microactuators are shown. These microactuators have been used in different systems and application areas due to their functionality, allowing the creation of new devices and systems, mainly microgrippers, which is the final goal of this work.

Table 2 shows some recently developed micromanipulator devices, as well as the type of microactuators used to carry out their operation.

In Section 2, the design, modeling, and simulation of the proposed microgripper are shown.

## 2. Materials and Methods

### 2.1. Design Concept and Simulation

Our starting point of design is the geometry of a V-shaped microactuator, with a pair of symmetrical arms powered by an electric potential. With the knowledge about the operation of the V-shaped microactuator and how to develop a geometry, whose characteristics allows us to improve the performance of at least one of its parameters, the following research questions were formulated: Q1: What would happen if the widths and beam lengths of the V-shaped beam microactuator were modified? Q2: Would the performance of the displacement or force parameters be improved?

To answer these questions, the proposed device was subjected to a parameterization analysis, applied to the width and length of its beams. The novel optimized microactuator is shown in Figure 1.

To validate the performance of the asymmetric microactuator, a novel microgripper is also proposed (Figure 2). It consists of two symmetrical arms, which are driven in their lower section by the asymmetrical actuator; on its upper section, a Z geometry was added, to increase the displacement of the jaws. In addition, an arrangement of compliant beams was included, which supports the arm.

Table 3 shows the electrical, mechanical, and thermal parameters of silicon (Si) for consideration in the simulation, using ANSYS™, as well as for the analytical models.

In Table 4, the dimensions of the microgripper elements are provided, which also will be considered in the simulations, as well as in the analytical models. It should be noted that the thickness of the structure was selected according to the SOI wafers used in the process in which they could be fabricated.

In Section 2.2 and Section 2.3, some models reported in the literature that investigate the thermal, mechanical, and electrical behavior of MEMS devices, applicable to the proposed microactuator and microgripper, are adjusted.

### 2.2. Modelling of the Asymmetric Actuator

#### 2.2.1. Displacement of the V-Shaped Actuator

The models for asymmetric microactuator were generated from others reported for the V-shaped actuator, or from well-known mechanical or electrical ones.

It is well-known that the force generated on a beam, when thermal energy is applied, that produces an increment in temperature can be calculated by
(1)$$F_{b}=A_{c}E\frac{\Delta L}{L_{0}}=A_{c}E\Delta T\alpha$$
where the length increment Δ*L* of a beam of length *L* is given by $L=L_{0}\Delta T\alpha$. *A_c_* is its cross-section area. The component of *F_b_* in the Y-axis is
(2)$$F_{by}=A_{c}E\Delta T\alpha sin\theta$$

The displacement of the V-shaped actuator is due to the Joule effect, creating a bending deformation of the beams [42]. In [41], a mathematical expression for the displacement was given, considering both lateral bending and axial deformation of the beams under a small deformation hypothesis, as a function of the thermal expansion of the beams.
(3)$$U_{y}=\frac{\Delta T\alpha Lsin\theta }{s^{2}+c^{2}\left(\frac{12I}{A_{c}L^{2}}\right)}$$
where Δ*T* is the average temperature increase, *c* = *cosθ* and *s* = *sinθ*, *L* is the length of the beam, and *I* is the second moment of inertia of the cross section.

#### 2.2.2. Displacement of the Asymmetric Actuator

The displacement equation proposed for the asymmetric actuator was developed using the stiffness matrix method, also known as the displacement method in frames. The main equation of this method is as follows [43]:(4)[K]{U}={F}+{R}
where [*K*] is the stiffness matrix of the frame, which is multiplied by the displacement vector {*U*}, equal to the sum of the forces {*F*} and reaction {*R*} vectors.

The analysis considered all the degrees of freedom at each end of both beams, which share the 4, 5, and 6 DOFs.

The stiffness matrix of the frame is given by the following equation [43]:(5)K=[123456EAL00−EAL00012EIL36EIL20−12EIL36EIL206EIL24EIL0−6EIL22EIL−EAL00EAL000−12EIL3−6EIL2012EIL3−6EIL206EIL22EIL0−6EIL24EIL]

The local displacements and forces are expressed as global ones. For beam 1,
(6)$$\left[\begin{array}{c} U_{Lx1}\\ U_{Ly2}\\ U_{L3}\\ U_{Lx4}\\ U_{Ly5}\\ U_{L6} \end{array}\right]=\left[\begin{array}{cccccc} c & s & 0 & 0 & 0 & 0\\ -s & c & 0 & 0 & 0 & 0\\ 0 & 0 & 1 & 0 & 0 & 0\\ 0 & 0 & 0 & c & s & 0\\ 0 & 0 & 0 & -s & c & 0\\ 0 & 0 & 0 & 0 & 0 & 1 \end{array}\right]\left[\begin{array}{c} U_{x1}\\ U_{y2}\\ U_{3}\\ U_{x4}\\ U_{y5}\\ U_{6} \end{array}\right]\;\mathrm{and}\;\left[\begin{array}{c} F_{x1}\\ F_{y2}\\ M_{3}\\ F_{x4}\\ F_{y1}\\ M_{3} \end{array}\right]=\left[\begin{array}{cccccc} c & -s & 0 & 0 & 0 & 0\\ s & c & 0 & 0 & 0 & 0\\ 0 & 0 & 1 & 0 & 0 & 0\\ 0 & 0 & 0 & c & -s & 0\\ 0 & 0 & 0 & s & c & 0\\ 0 & 0 & 0 & 0 & 0 & 1 \end{array}\right]\left[\begin{array}{c} F_{Lx1}\\ F_{Ly2}\\ M_{L3}\\ F_{Lx4}\\ F_{Ly5}\\ M_{L6} \end{array}\right]$$

Equation (7) is obtained from substituting Equations (5) and (6) into Equation (4) and is as follows:(7)$$\left[\begin{array}{cccccc} 1 & 2 & 3 & 4 & 5 & 6\\ \left(c^{2}M+s^{2}N\right) & cs\left(M-N\right) & -sm & -(c^{2}M-s^{2}N) & cs\left(-M+N\right) & -sm\\ cs\left(M-N\right) & \left(s^{2}M+c^{2}N\right) & cm & cs\left(-M+N\right) & \left(-s^{2}M-c^{2}N\right) & cm\\ -sm & c^{2}m & n & sm & -cm & r\\ \left(-c^{2}M-s^{2}N\right) & cs\left(-M+N\right) & sm & \left(c^{2}M+s^{2}N\right) & cs\left(M-N\right) & sm\\ cs\left(-M+N\right) & \left(-s^{2}M-c^{2}N\right) & -cm & cs\left(M-N\right) & \left(s^{2}M+c^{2}N\right) & -cm\\ -sm & sm & r & sm & -cm & n \end{array}\right]\left[\begin{array}{c} U_{x1}\\ U_{y2}\\ U_{3}\\ U_{x4}\\ U_{y5}\\ U_{6} \end{array}\right]\;=\;\left[\begin{array}{c} F_{x1}\\ F_{y2}\\ M_{3}\\ F_{x4}\\ F_{y5}\\ M_{6} \end{array}\right]+\left[\begin{array}{c} R_{1}\\ R_{2}\\ R_{3}\\ R_{4}\\ R_{5}\\ R_{6} \end{array}\right]$$

Similarly, the following equation is obtained for beam 2:(8)$$\left[\begin{array}{cccccc} 4 & 5 & 6 & 7 & 8 & 9\\ \left(c^{2}M+s^{2}N\right) & cs\left(M-N\right) & -sm & -(c^{2}M-s^{2}N) & cs\left(-M+N\right) & -sm\\ cs\left(M-N\right) & \left(s^{2}M+c^{2}N\right) & cm & cs\left(-M+N\right) & \left(-s^{2}M-c^{2}N\right) & cm\\ -sm & c^{2}m & n & sm & -cm & r\\ \left(-c^{2}M-s^{2}N\right) & cs\left(-M+N\right) & sm & \left(c^{2}M+s^{2}N\right) & cs\left(M-N\right) & sm\\ cs\left(-M+N\right) & \left(-s^{2}M-c^{2}N\right) & -cm & cs\left(M-N\right) & \left(s^{2}M+c^{2}N\right) & -cm\\ -sm & sm & r & sm & -cm & n \end{array}\right]\left[\begin{array}{c} U_{x4}\\ U_{y5}\\ U_{6}\\ U_{x7}\\ U_{y8}\\ U_{9} \end{array}\right]\;=\;\left[\begin{array}{c} F_{x4}\\ F_{y5}\\ M_{6}\\ F_{x7}\\ F_{y8}\\ M_{9} \end{array}\right]+\left[\begin{array}{c} R_{4}\\ R_{5}\\ R_{6}\\ R_{7}\\ R_{8}\\ R_{9} \end{array}\right]$$

For the sake of simplicity, the elements of the stiffness matrixes are denoted by *M* = (*EAc*/*L*), *N* = (*12 EI*/*L*^3^), *m* = (*6 E*/*L*^2^), *n* = (*4 EI*/*L*) *y r* = (*2 EI*/*L*), where *A* is the corresponding cross-section area of the corresponding beam, *c* = *cosθ*, and *s* = *sinθ*, with *θ* in grades.

Equations (7) and (8) were added to obtain the total system equation. As a first approximation, the assumptions and conditions that were used in [41] for the symmetric chevron were considered. *U_x_* = 0, i.e., there is no displacement in the X-axis, *U_y_* ≠ 0, meaning there is displacement in the *Y*-axis and *R_y_* = *R_x_* = *0*, and *M*_3_ = *0*, meaning there are no reactions or moments. The only displacement that will be generated is in DOF 5, in the *Y*-axis, since DOFs 2 and 8 are embedded to the anchors and the displacement matrix is given by
(9)$$\left[\begin{array}{c} F_{x1}\\ F_{y2}\\ M_{3}\\ {\left(F_{x4}\right)}_{1}+{\left(F_{x4}\right)}_{2}\\ {\left(F_{y5}\right)}_{1\;}+{\left(F_{y5}\right)}_{2\;}\\ {\left(M_{6}\right)}_{1}+{\left(M_{6}\right)}_{2}\\ F_{x7}\\ F_{y8}\\ M_{9} \end{array}\right]+\left[\begin{array}{c} R_{1}\\ 0\\ R_{3}\\ {\left(R_{4}\right)}_{1}+{\left(R_{4}\right)}_{2}\\ 0\\ {\left(R_{6}\right)}_{1}+{\left(R_{6}\right)}_{1}\\ R_{7}\\ 0\\ R_{9} \end{array}\right]=\left[\begin{array}{c} \left(cs\left(-M+N\right)\right)\;U_{5}\\ \left(s^{2}M-c^{2}N\right)U_{5}\\ \left(c^{2}m\right)U_{5}\\ \left({\left[cs\left(M-N\right)\right]}_{1}+{\left(cs\left(M-N\right)\right)}_{2}\right)U_{5}\\ \left({\left(s^{2}M+c^{2}N\right)}_{1}+{\left(s^{2}M-c^{2}N\right)}_{2}\right)U_{5}\\ \left({\left(-cm\right)}_{1}+{\left(c^{2}m\right)}_{2}\right)U_{5}\\ {\left(cs\left(-M+N\right)\right)}_{2}U_{5}\\ {\left(-s^{2}M-c^{2}N\right)}_{2}U_{5}\\ {\left(sm\right)}_{2}U_{5} \end{array}\right]$$

Considering only the expression for the DOF 5, Equation (9) is reduced to
(10)$${\left(F_{y5}\right)}_{1\;}+{\left(F_{y5}\right)}_{2\;}=\left({\left(s^{2}M+c^{2}N\right)}_{1}+{\left(s^{2}M-c^{2}N\right)}_{2}\right)U_{5}\to \;U_{5}=\frac{{\left(F_{y5}\right)}_{1\;}+{\left(F_{y5}\right)}_{2\;}}{\left({\left(s^{2}M+c^{2}N\right)}_{1}+{\left(s^{2}M-c^{2}N\right)}_{2}\right)}$$

By substituting Equation (2), and considering *U*_5_ = *U_y_*, and the corresponding subindexes 1 and 2, for the case of the geometrical characteristics of beam 1 and 2, respectively, *U_y_* is obtained as
(11)$$U_{y}=\frac{\alpha \Delta Tsin\theta \left(A_{1}+A_{2}\right)}{s^{2}\left(\frac{A_{1}}{L_{1}}+\frac{A_{2}}{L_{2}}\right)+12c^{2}\left(\frac{I_{1}}{L_{1}^{3}}+\frac{I_{2}}{L_{2}^{3}}\right)}$$
where *A*_1_ and *A*_2_ are the cross-section areas, *L*_1_ and *L*_2_ the beam lengths, *I*_1_ and *I*_2_ are the inertia moments in (m^4^) of beams 1 and 2, respectively. $I_{1}=\frac{tw_{1}^{3}}{12}$, and $I_{2}=\frac{tw_{2}^{3}}{12}$.

#### 2.2.3. Force of V-Shaped Actuator

In [25], the stiffness of the V-shaped clamped thermal beams can be represented by the following equation:(12)$$k=\frac{2NE\left(12Icos^{2}\theta +AL^{2}sin^{2}\theta \right)}{L^{3}}$$
where *N* is number of the pairs of beams. The other variables were previously defined.

The force due to a pair of beams can be obtained as *F_y_* = *kU_y_*, with *k* and *U_y_* given by Equation (12), with *N* = *1*, and Equation (3), respectively.

#### 2.2.4. Force of Asymmetric Actuator

From Equation (4), and considering the different cross-section areas (*A*_1_ and *A*_2_) and lengths (*L*_1_ and *L*_2_) of beams, the equivalent stiffness coefficient can be given by:(13)$$k=k_{1}+k_{2}$$
with
(14)$$k_{1}=\frac{E\left(12I_{1}cos^{2}\theta +AL_{1}^{2}sin^{2}\theta \right)}{L_{1}^{3}}\;\mathrm{and}\;k_{2}=\frac{E\left(12I_{2}cos^{2}\theta +AL_{2}^{2}sin^{2}\theta \right)}{L_{2}^{3}}$$
*k_1_* and *k_2_* were obtained from Equation (12). The force in the Y direction can be calculated as *F_y_* = *kU_y_*, with *k* and *U_y_* given by Equations (11) and (13), respectively.

#### 2.2.5. Electric Modelling of the Asymmetrical Microactuator

The electrical resistance of the proposed asymmetrical beam actuator (Figure 3), according to the notation given in (Figure 1), can be given by the following equation:(15)$$R_{e}=\rho \left(\frac{2L_{a}}{A_{a}}+\frac{L_{1}}{A_{1}}+\frac{2L_{2}}{A_{2}}\right)$$
where *L_a_* and *A_a_* are the length and cross-section area of the anchor, respectively, and $L_{2}=2L_{1}$. Electric current can be calculated from Equation (6), as *I* = *V*/*R_e_*.

*R_e_* = 248.57 Ω is obtained by Expression (6), considering *ρ* = 1.5 × 10^−4^ Ω × m.

### 2.3. Electrical Modeling of Microgripper

A half of the microgripper was considered (Figure 4) for the resistive analysis of the microgripper (Figure 5). The values of the resistances *R*_1_ to *R*_13_ are calculated by the well-known relation $R=\rho \frac{L}{A}$, where *ρ* is the resistivity, *L* and *A* are the length and the corresponding cross-section of the beam, respectively. The geometric sizes of the elements with *R*_1_ to *R*_13_ are given in Table 5. We considered *ρ* = 1.5 × 10^−4^ Ω*m, and *t* = 70 µm.

The resistive equivalent of one half of the microgripper, *R_h_*, is obtained by performing the corresponding simplifications of the resistive circuit shown in Figure 5a, considering two transformations, the first one from delta to star, and the second one from star to delta. Therefore, the total resistance, *R_T_*, of the microgripper is calculated as the parallel of the two resistances *R_h_* as follows:(16)$$R_{T}=\frac{R_{h}}{2}=\frac{R_{1}+R_{12}+R_{13}+\frac{R_{10}R_{11}}{R_{9}+R_{10}+R_{11}}}{2}\;\;\;\;\;\;\;\;\;\;\;\;\;\;\;\;\;\;\;\;\;\;+\frac{\left(\frac{\mathrm{num}1}{R_{4}}\right)\left[\frac{\left(\frac{R_{3}}{R_{5}+R_{6}+R_{7}+\frac{R_{9}R_{11}}{R_{9}+R_{10}+R_{11}}}\right)}{R_{3}+\frac{\mathrm{num}1}{R_{5}+R_{6}+R_{7}+\frac{R_{9}R_{11}}{R_{9}+R_{10}+R_{11}}}}+\frac{\left(R_{8}+\frac{R_{9}R_{10}}{R_{9}+R_{10}+R_{11}}\right)\frac{1}{R_{2}}}{R_{8}+\frac{R_{9}R_{10}}{R_{9}+R_{10}+R_{11}}+\frac{\mathrm{num}1}{R_{2}}}\right]}{2\left[\frac{1}{R_{4}}+\frac{\left(\frac{R_{3}}{R_{5}+R_{6}+R_{7}+\frac{R_{9}R_{11}}{R_{9}+R_{10}+R_{11}}}\right)}{R_{3}+\frac{\mathrm{num}1}{R_{5}+R_{6}+R_{7\;}+\frac{R_{9}R_{11}}{R_{9}+R_{10}+R_{11}}}}+\frac{\left(R_{8}+\frac{R_{9}R_{10}}{R_{9}+R_{10}+R_{11}}\right)\frac{1}{R_{2}}}{R_{8}+\frac{R_{9}R_{10}}{R_{9}+R_{10}+R_{11}}+\frac{\mathrm{num}1}{R_{2}}}\right]}$$
with
$$\mathrm{num}1=\frac{\left(R_{2}+R_{4}\right)\left(R_{5}+R_{6}+R_{7}+\frac{R_{9}R_{11}}{R_{9}+R_{10}+R_{11}}\right)+R_{2}R_{4}}{R_{4}}$$
with *ρ* = 1.5 × 10^−4^ Ω × m, *Re* = 41.74 Ω.

## 3. Results

### 3.1. Multiphysics FEM Model of Asymmetric Microactuator

To choose the geometry that provides the larger force, the simulation of the asymmetric actuator was performed, considering the following two cases: microactuator 1, where the arms have different lengths and equal widths and microactuator 2, with arms of different lengths and widths. Table 6 shows the details of the multiphysics elements considered in the corresponding simulations.

The results obtained when the actuators were fed with 2 V are shown in Figure 6.

With regard to the boundary conditions for the device simulated in Ansys Workbench, the actuator was fed with 2 V, and the room temperature was considered *as T_a_* = 22 °C. As part of the device operating conditions, the pads were fixed. The left pad was set to a temperature of 22 °C, and its electric potential was set to 0 V. The right pad was set to 2 V. It should be noted that the simulations of microactuators 1 and 2 were performed with the same boundary conditions.

For comparison, a V-shaped microactuator was developed with a length of 600 µm, the same total length, and width of 5 µm, and with the same shaft dimensions of the asymmetric actuators.

To numerically determine the temperature distribution and the displacement of the devices with respect to their length, a simulation trajectory was created along each device (from pad to pad). The results are shown in Figure 7. They allowed us to choose the most suitable asymmetric microactuator to apply to the microgripper.

The simulation results of force and equivalent von Mises stress for microactuators 1, 2, and the V-shaped microactuator are given in Figure 8.

From the graphs shown in Figure 7a, it is observed that, with respect to actuator 2, the temperature distribution shows a peak at the thin arm length of ≈350 µm, from which the temperature tends to remain constant, in the range between ≈105 °C and 120 °C. In the case of actuator 1, an increasing trend is also observed, but smaller than in the case of microactuator 2; there is also a tendency to stabilize in the same range as microactuator 2. In the case of the V-shaped microactuator, a similar trend to microactuator 1 is observed, but with a greater convergence of values. In summary, the highest temperature values are obtained with microactuator 2.

From Figure 7b, a greater symmetry of the displacement is observed in the case of the V-shaped microactuator, followed by microactuator 1. In both cases, the maximum of the curves is localized at its midpoint, with a slight shift to the left in the case of actuator 1. Of the three actuators, actuator 1 achieves the largest displacement (6.69 µm), while actuator 2 achieves a maximum displacement of 4.8 µm, at the end of its thin beam.

Regarding the force, from Figure 8a, we can observe the increasing tendency of the three actuators, with higher values for actuator 2, reaching a maximum of 24 mN, while actuator 1 reaches a value of 5.4 mN, when they are fed with 2 V. This remarkable increase in force was the reason for choosing microactuator 2.

For all actuators, the stress values are below the ultimate stress for Si, with larger variations in the case of microactuator 1, but with larger values in the thin beam of microactuator 2. The highest stress values occur in actuator 1 and the V-shaped actuator, very close to the junction point of their corresponding beams with the pad, where the positive potential is applied (Figure 8b,c).

The simulation results for all the actuators analyzed are summarized in Table 7, where the results for a conventional V-shaped actuator are also included.

It can be observed from Table 7 that the increase in force of actuator 2 is superior to the others, exceeding the V-shaped force by 370.38%, but with a 29.8% decrease in displacement. The force increase is the reason for choosing it to actuate the proposed micro gripper.

In Figure 9, the results of the simulations of equivalent von Mises stress are shown. The maximum value is given for each actuator in 3 points of interest. In Table 8, these results are summarized for them.

As it can be observed in Figure 9 and Table 8, in all cases shown, the equivalent von Mises stress values are smaller than the ultimate stress for Si. The location of the maximum stress is also similar, near to the anchor where the voltage source is applied.

The analytical result of the displacement for actuator 2 using Equation (11) is 3.29 µm, with an error of 31%, in relation to the numerical values, while F_y_ = 17.3 mN with an error of 28% and the the stiffness calculated using Equation (13) was 5250 N/m, with an error of 0.05%. It is necessary to mention that in simulation, thermal convection was used, which could increase the errors. Similar errors were observed for the case of the V-shaped actuator, where displacement has an error of 29%. In addition, in this first approach to the displacement calculation of the asymmetric actuator, no moments or reactions were considered.

### 3.2. Multiphysics FEM Model of Microgripper

The proposed microgripper was simulated in ANSYS™ using Si, with the microactuator 2, with and without damping elements, and with the V-shaped microactuator (Figure 10) to compare the displacement between its jaws. It was considered to maintain the reduced total area. The technical details are shown in Table 9.

As can be observed from Figure 10, the displacement of the microgripper is smaller in the case without damping elements. The smallest displacement corresponds to the microgripper actuated with the V-shaped actuator.

*MWD = microgripper with damping elements; *MWoutD = microgripper without damping elements; *MWVS = microgripper with V-shaped actuator.

The simulation results of the microgripper with actuator 2, with and without damping elements, and with the V-shaped actuator are shown in Figure 11, considering their relationship of distribution of temperature to the length of the microactuators, the force to the voltage applied, and displacement to the length of the jaws. The highest values of these parameters correspond to the microgripper actuated by the micro actuator 2. Table 10 summarizes the results at 2 V.

Regarding the displacement in the Z-axis, *U_z_* = 0.023 µm for the microgripper with damping elements, and without them, *U_z_* = 0.022 µm. In both cases, these values are small compared to the displacement in the other axis.

From Table 10, it is observed that MWD has the larger displacement and force, as expected. It also has the larger natural frequency, notably improving its response.

The equivalent von Mises stress results are shown in Figure 12 and Table 11 in the specified points of interest to observe the microgrippers’ performance and the location of the maximum point of stress.

According to Figure 12, the maximum values for both cases are located near the junction point between the beam and the anchorage where the stress source is applied. The maximum value of MWD is 68.65 MPa, which is smaller than the maximum value obtained for MWoutD. This is another advantage of the use of damping elements.

In Figure 13, the simulation results of current for the actuator 2 and the microgripper, with and without damping elements, are given. At 2 V, the calculated current value for MWD is 47.9 mA, and 8.04 mA for the microactuator 2. The error for MWD´s current, compared with the numeric approach, is 13.8%, and for the actuator, it is only 0.0047%, which validates our electric models.

## 4. Discussion

From the results given in the previous section for the microactuator proposed, Q1 can be answered by mentioning that when the widths and lengths of the arms of a V-shaped actuator are modified, considering the proportions for the microactuator 2 (*L*_2_ = *2 L*_1_, and *W_2_* = *5 W*_1_), and applying 2 V, the temperature increases nonlinearly until approximately 100 °C in the short beam at ≈350 µm, and after, the growth is slower with a tendency to remain constant, in the range between ≈105 °C and 120 °C (Figure 7a). This actuator was compared with actuator 1 (*L*_2_ = *2 L*_1_, and *W*_2_ = *W*_1_) and the V-shaped actuator (*L* = *3 L*_1_/*2*, *W* = *W*_1_). In all cases, the temperature in the short beam of microactuator 2 was higher, which is a disadvantage.

The displacement decreases from 6.23 µm to 4.8 µm (Figure 7b), that is, a reduction of 29.8% compared with the case of the V-shaped microactuator; however, the force goes from 5.1 mN to 24 mN (Figure 8a), which is equivalent to an increment of 370.58%. The equivalent von Mises stress increases in the microactuator 2, with the largest values at the end of the thin beam (74.2 MPa), near to the anchor where the voltage source was applied, but with values under the ultimate stress of Si (250 MPa) (Figure 8b and Figure 9). The 3D graphical comparison shown in Figure 8c allows us to observe the highest stress values for microactuator 1, and the V-shaped microactuator at the end of the beams near to the corresponding anchors where the voltage is applied, as well as the more punctual locations.

The answer to Q2 is that the improved parameter is the force, which, for our application, is very useful.

For the case of the novel microgripper, which is normally closed, with an initial aperture of 50 µm, it was implemented with the analyzed microactuators, maintaining a similar total area. A disadvantage of using microactuator 2 is given by its temperature increment, compared to the case when a V-shaped actuator is used and on average increases from 45° to 72°, with an increment of 60% (Figure 11a). The temperature, however, is not excessively high, so there is a range of microobjects that can be clamped.

From the data given in Table 10, at 2 V, actuator 2 with damping elements (MWD) has an aperture of 4.85 µm, which represents an increment of 370% compared when the V-shaped actuator. The force showed an increment of 74.80%, reaching a value of 73.61 mN. The natural frequency had an increment of 234%, with *f_n_* = 37.994 kHz. Its maximum equivalent von Mises stress was 68.65 MPa, a lower value than the ultimate stress for Si (Table 11). The performance parameters of MWD also have larger values than the case of MWoutD (Table 10 and Table 11).

At 2 V, the calculated current value for the MWD is 47.9 mA, and 8.04 mA for the microactuator 2. The errors relative to the numerical analysis are 13.8%, and 0.0047% (Figure 13), which validates our electrical models.

The largest diameter that could be clamped is 54.85 µm. The initial aperture can be easily modified, increasing the versatility of the use of this microgripper. Because of the increase in force up to 73.61 mN, it can support masses up to 7.5 mg.

Due to the high force, some immediate applications could be found in the research of materials or microdevices for observation, or experimental tests of stress, among others. Currently, several microobjects are under development for diverse areas, for example, micro components for microdevices, microwires or microfibers for RF or optic communications, and micro assembling for micro manufacture.

The microgripper has a relatively high gripping force with a compact design, making it suitable for potential applications as part of microinjection or drilling systems that are required in several fields, such as micromanufacture and assembling processes. However, for this possible application, future intensive research is required.

With the developed theoretical approaches and the numerical results, the performance of the proposed microstructures has been demonstrated. More complex theoretical models of the microactuator and the microgripper could be developed in the future. The fabrication of the structure and the corresponding experimental tests could also be performed.

## 5. Conclusions

In the novel and optimized microgripper, the use of a new asymmetric electrothermal microactuator is remarkable. The microgripper is characterized by its Z-beam shape and damping elements. In the case of the microactuator based on a V-shaped actuator, only two beams are needed, with different lengths and widths, and a small prebending angle (1°, in relation to the horizontal angle) and in comparison with the V-shaped microactuator of two beams of equal total length, the increment in force is 3.7 times, but the decrement in displacement is 29.8%.

Benefiting from the new actuator design concept, the total size of the entire gripper is only 2.068 mm × 0.527 mm × 0.07 mm. At the same time, by increasing the driving voltage applied to the asymmetric thermal microactuator to 2 V, the force increases until to 73.61 mN and the displacement up to 4.8 µm. With the implementation of the asymmetric microactuator, the stress level is low and its highest value on the arrow is 37.93 MPa, while the ultimate tensile stress of silicon is 250 MPa. To avoid collapsing and nonlinear displacements in the microgripper, the implementation of the damping elements is included in the microactuator design, which increases the force, displacement, and frequency range.

Our simulation and analytical results verify the feasibility of the MEMS gripper design. This geometry can be used in other fabrication methods, such as the numerical control machine, which may be considered in future work.

## Figures and Tables

**Figure 1 micromachines-13-01460-f001:**
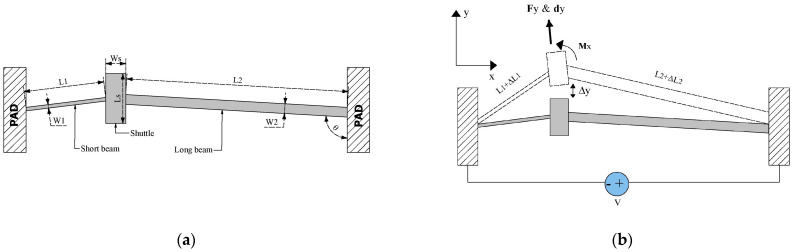
(**a**) Elements and geometric sizes, and (**b**) operating conditions of the asymmetric microactuator.

**Figure 2 micromachines-13-01460-f002:**
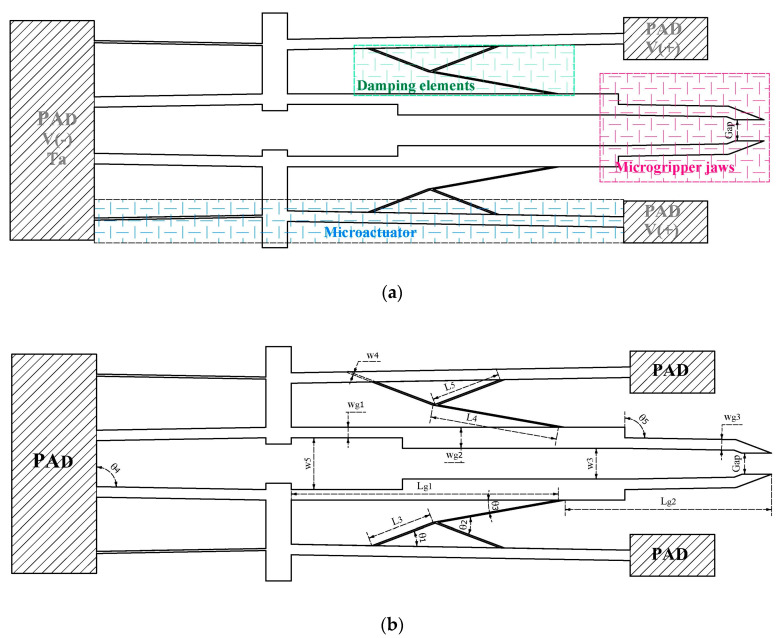
(**a**) Elements and (**b**) geometric sizes of the novel electrothermal microgripper.

**Figure 3 micromachines-13-01460-f003:**
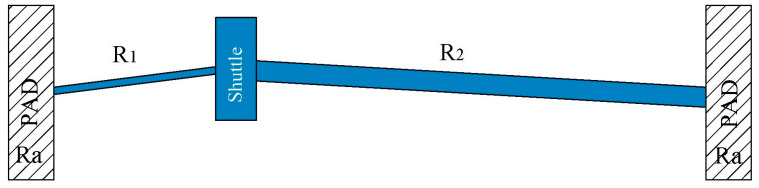
Resistances of asymmetric actuator.

**Figure 4 micromachines-13-01460-f004:**
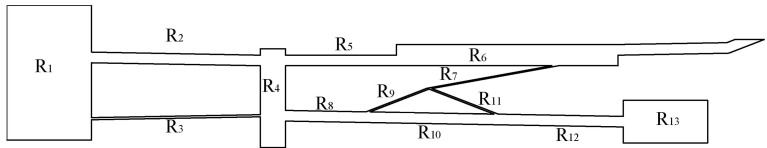
A half of the microgripper.

**Figure 5 micromachines-13-01460-f005:**
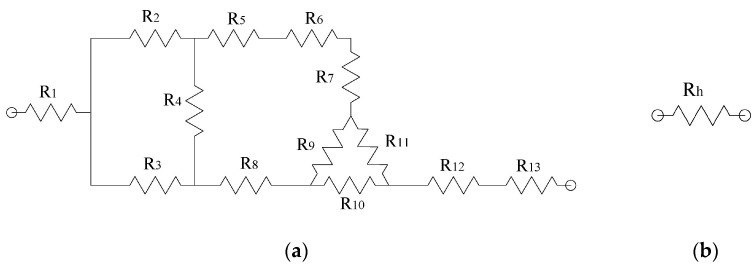
(**a**) Resistive equivalent circuit of one half of the microgripper with damping elements. (**b**) Equivalent resistance.

**Figure 6 micromachines-13-01460-f006:**
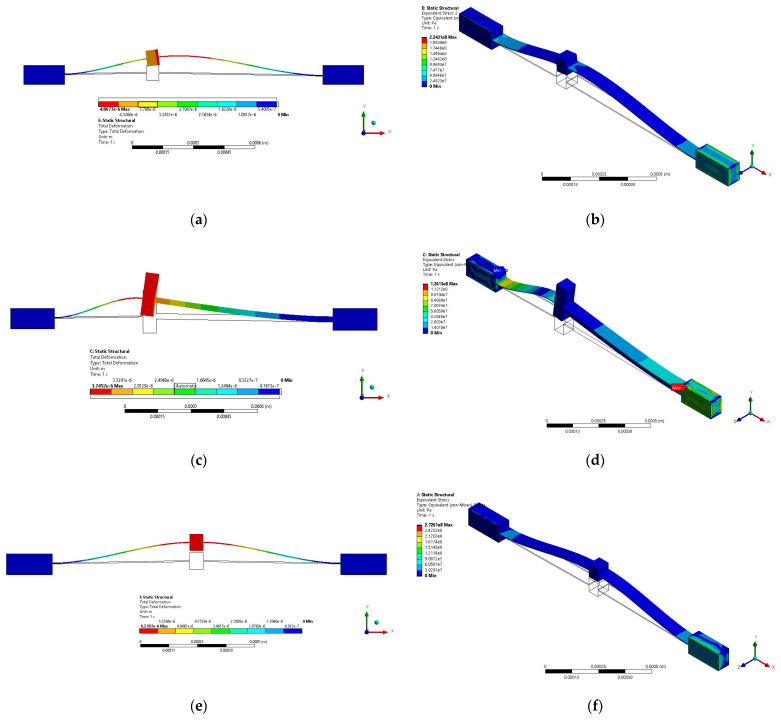
Simulation results of (**a**) deformation and (**b**) equivalent von Mises stress of microactuator 1. (**c**) Deformation and (**d**) equivalent von Mises stress of microactuator 2. (**e**) Deformation and (**f**) equivalent von Mises stress of V-shaped microactuator.

**Figure 7 micromachines-13-01460-f007:**
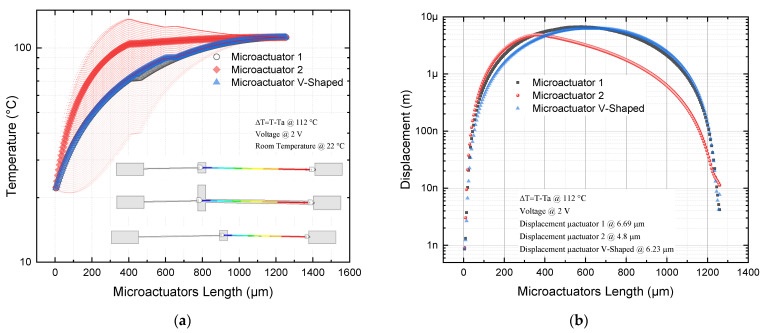
(**a**) Temperature and (**b**) displacement of microactuators 1, 2, and V-shaped.

**Figure 8 micromachines-13-01460-f008:**
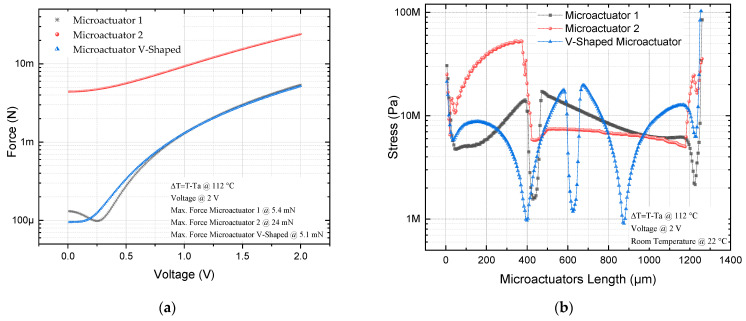
(**a**) Force, and (**b**) 2D and (**c**) 3D graph of equivalent von Mises stress microactuators 1, 2, and V-shaped.

**Figure 9 micromachines-13-01460-f009:**
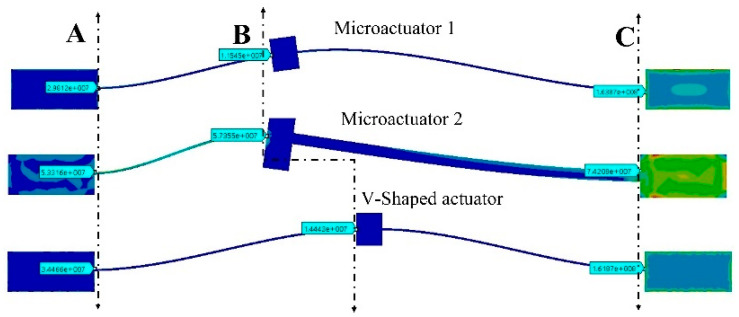
Results of equivalent von Mises stress for the microactuators considered. A represents the pad fed with 0 V, B corresponds to the shuttle location of each actuator. C indicates the pad fed at 2 V. In A, B and C the von Mises stress values are shown.

**Figure 10 micromachines-13-01460-f010:**
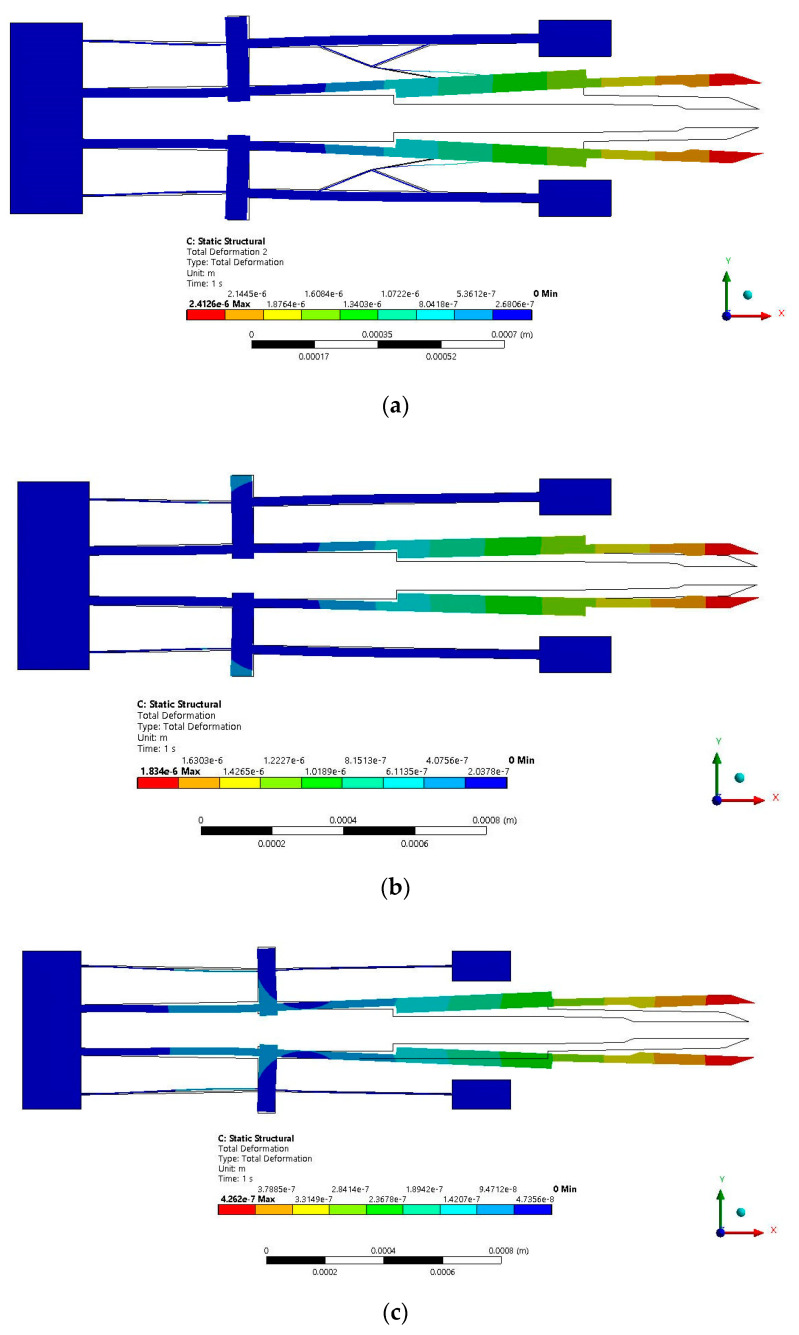
Displacement of microgripper (**a**) with and (**b**) without damping elements. (**c**) Displacement of microgripper with a V-shaped actuator.

**Figure 11 micromachines-13-01460-f011:**
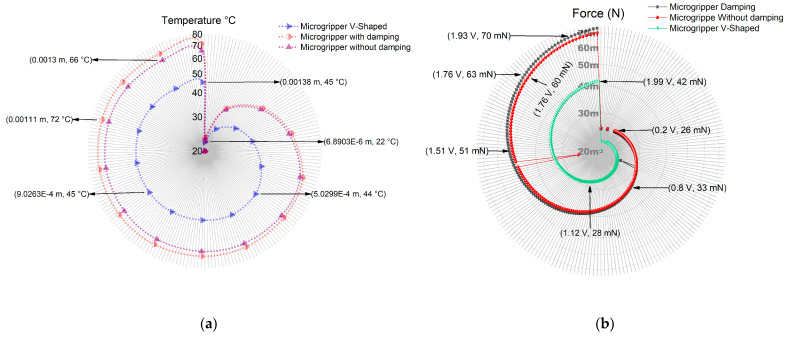
(**a**) Temperature distribution vs. length, (**b**) force vs. applied voltage, and (**c**) displacement vs. jaw length for the micro gripper with actuator 2, with and without damping elements, and with the V-shaped actuator.

**Figure 12 micromachines-13-01460-f012:**
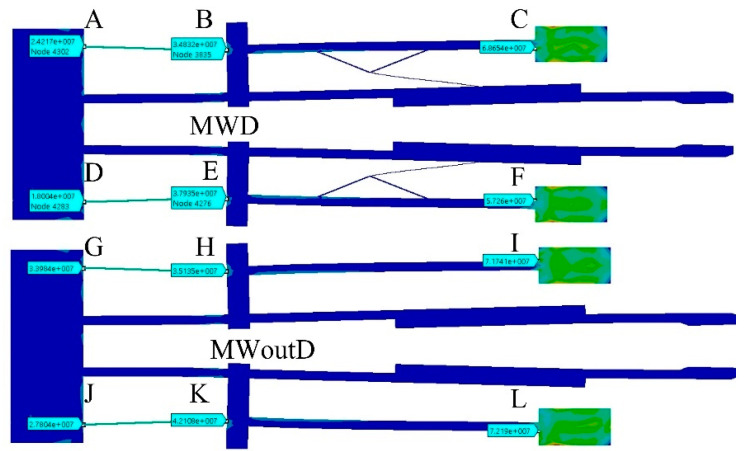
Equivalent von Mises stress distribution for MWD, and MWoutD. A, D, G, and J represent the pad fed with 0 V. B, E, H, and K correspond to the shuttle location of each actuator. C, F, I, and L indicate the pad fed at 2 V.

**Figure 13 micromachines-13-01460-f013:**
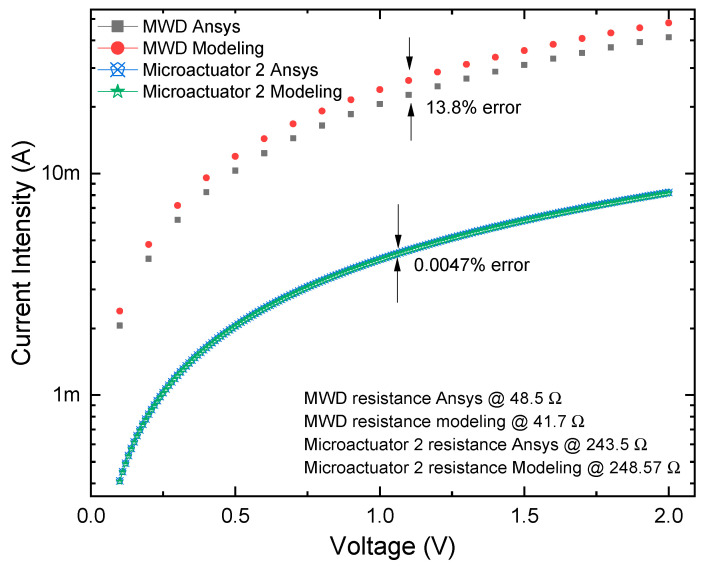
Behavior of current vs. voltage by numeric and modeling results.

**Table 1 micromachines-13-01460-t001:** State of the art of recent V- and Z-shaped beam microactuators, and some of their fundamental parameters.

Ref.	Microactuator Type	Structural Material	Number of Pair of Beams	Inclination Angle	Dimensions (µm)	Software for Simulation	Displacement (µm)	Force (µN)	Stiffness (N/m)
[3]	V-shaped	Poly-Si	1	NA	600 × 100 × 20	Ansys™	NA	NA	NA
[12]	Planar magnetostrictive	Ni	6	4°	4 × 2 × 0.4	Comsol™	10.2	NA	5.56
[24]	Chevron thermal	Al	4	10°	510 × 335 × 10	Comsol™	10.94	NA	NA
[25]	V-shaped	Si	10	2°	≈1500 × 300 × 30	Ansys™	70	NA	NA
[26]	V-shaped	Poly-Si	10	10°	≈600 × 400 × 10	Comsol™	0.6	NA	NA
[28]	Z-shaped	Si	2	NA	412 × 60 × 10	Ansys™	0.2107	NA	NA
[30]	V-shaped	Ni	1	0.5°	≈1.5 × 12 × 21	Ansys™	≈50	1000	NA
[31]	Z-shaped	Si	2	10°	≈176 × 88 × 10	Ansys™	0.750	30–40	NA
[32]	V-shaped	Poly-Si	3	NA	≈600 × 4 × 6.95	Abaqus™	≈5	≈400	NA

Note: not available (NA).

**Table 2 micromachines-13-01460-t002:** State of the art of recent microgrippers, their actuators, and fundamental parameters.

Ref.	Microgripper Type	Microactuator Type	Structural Material	Simulated or Fabricated	Dimensions (µm)	Displacement of Tips (µm)	Initial Gap (µm)	Stress Max (kPa)	Force on Tips (µN)	Stiffness (N/m)
[33]	Electrothermal	U-beam	Si	Fabricated	≈375 × 200 × 60	≈11 at 9 V	≈15	NA	NA	NA
[34]	Electrostatic and piezoelectric	Two fully clamped symmetrically microbeams	Si and PZT	Simulated	≈600 × 600 × NA	2 at 18 V	≈2	≈0.156	NA	NA
[35]	Electrothermal	Chevron	Poly-Si	Simulated	≈1000 × 900 × 10	19.2 at 1 V	100	470	0–17,000	NA
[36]	Electrothermal	Z-shaped	Poly-Si	Simulated	≈2680 × 2750 × 50	80 at 6 V	100	Na	6575	263 × 10^−6^
[37]	Electrothermal	U-shaped	Poly-Si	Fabricated	≈280 × 100 × NA	9.1 at 14 V	20	104	36 to 14 V	4.05
[38]	Electrothermal	U-shaped	Poly-Si	Fabricated	≈1000 × 210 × 2	19.6 at 5 V	5	ND	0.011	NA
[39]	Electrothermal	V-shaped	SU-8	Simulated	≈1650 × 800 × 9.85	11 at 80 mV	ND	22	231	NA
[40]	Electrothermal	Z-shaped	SU-8	Fabricated	≈1300 × 500 × 20	80 at 0.4 V	203.8	NA	26.3	NA

**Table 3 micromachines-13-01460-t003:** Mechanical and electrical parameters of Si [9,24,33,41].

Parameters	Silicon Values
Density ρ (kg/m^3^)	2329
Thermal expansion coefficient, α (C^−1^)	2.568 × 10^−6^
Young’s modulus, *E* (GPa)	130.1
Poisson’s ratio, ν	0.33
Isotropic thermal conductivity, *κ* (W/m °C)	148
Isotropic resistivity, $\rho_{0}$ (Ω × m)	0.00015
Average heat transfer coefficient, *h* (W/m^2^K)	25
Ultimate strength, (MPa)	250
Convection coefficient (W/m^2^ °C)	25

**Table 4 micromachines-13-01460-t004:** Geometrical parameters of the microgripper proposed.

Element Description	Dimensions (µm)	Element Description	Dimensions (µm)
Length of the short and thin beam of the microactuator (*L*_1_)	400	Gripper length from shaft to damping elements 1 (*L_g_*_1_)	631
Length of long and thick beam length of the microactuator (*L*_2_ = 2 × *L*_1_)	800	Gripper length from damping elements to jaw 2 (*L_g_*_2_)	770
Width of the short and thin beam of the microactuator (*w*_1_)	5	Width of the base beam of gripper 1 (*w_g_*_1_)	25
Width of long and thick beam length of the microactuator (*w*_2_)	25	Width of the Z section of gripper 2 (*w_g_*_2_)	50
Length of shuttle (*Ls*)	192.5	Width of the base of jaw 3 (*w_g_*_3_)	25
Width of shuttle (*Ws*)	60	Thickness of the structure (*t*)	70
Length of damping beam 1 (*L*_3_)	154.5	Gap (initial aperture between jaws)	50
Length of damping beam 2 (*L*_4_)	301	Pre-bending angle of the microactuator beams (*θ*)	91°
Length of damping beam 3 (*L*_5_)	170.5	Pre-bending angle of the damping beam 2 (*θ*_2_)	22°
Width of upper gap between gripper arms (*w*_3_)	78.5	Pre-bending angle of the damping beam 3 (*θ*_3_)	31°
Width of damping beams (*w*_4_)	9.5	Pre-bending angle between the beam base of the gripper and the pad (*θ*_4_)	80°

**Table 5 micromachines-13-01460-t005:** Geometric sizes of the elements with *R_1_* to *R_13_*.

Resistance	Length (µm)	Width (µm)	Resistance	Length (µm)	Width (µm)
*R_1_*	*200*	*263.72*	*R_8_*	*192*	*25*
*R_2_*	*400*	*25*	*R_9_*	*154.6*	*3.5*
*R_3_*	*400*	*5*	*R_10_*	*293.56*	*25*
*R_4_*	*232.5*	*60*	*R_11_*	*170.44*	*3.57*
*R_5_*	*400*	*25*	*R_12_*	*300*	*25*
*R_6_*	*240.92*	*50*	*R_13_*	*200*	*100*
*R_7_*	*300.7*	*1.75*			

**Table 6 micromachines-13-01460-t006:** Technical details about FEA in Ansys Workbench for microactuators 1 and 2, and the V-shaped microactuator.

Device	Solver Target	Element Type/Mesh/Number of DOF	Face Sizing with Element Size	Inflation			Convergence		Total Mass (kg)
				Transition Ratio	Max. Layers	Growth Rate	No. of Total Nodes	No. of Total Elements	
Microactuator 1	Mechanical APDL	SOLID 187/refinement controlled program	Default	0.272	5	1.2	3941	1749	0.7776 × 10^−8^
Microactuator 2							3003	1324	1.176 × 10^−8^
V-shaped microactuator							26,237	12507	7.77 × 10^−9^

**Table 7 micromachines-13-01460-t007:** Comparison of performance parameters of microactuators 1 and 2, and the V-shaped microactuator obtained by simulation in ANSYS™.

Device	Displacement @ 2 V (µm)	Force at 2 V (mN)	Stiffness (N/m)
Microactuator 1	6.69	5.4	807.17
Microactuator 2	4.8	24	5000
V-shaped actuator	6.23	5.1	818.620

**Table 8 micromachines-13-01460-t008:** Equivalent von Mises voltage results for the considered microactuators.

Device	Maximum Von Misses Stress (MPa)		
	Point A	Point B	Point C
Microactuator 1	29.8	11.5	163.8
Microactuator 2	53.3	57.3	74.2
V-shaped actuator	34.4	14.4	161.8

**Table 9 micromachines-13-01460-t009:** Technical details about FEA in Ansys Workbench for microgripper simulation.

Device	Solver Target	Physics Type and Analysis Type	Element Type/Mesh/Number of DOF	Inflation			Convergence		Total Mass (kg)
				Transition Ratio	Max. Layers	Growth Rate	No. of Total Nodes	No. of Total Elements	
*MWD	Mechanical APDL	Electric -> steady-state Thermal-electric conduction (1)	Solid187/refinement/39356	0.272	5	1.2	20,223	10877	0.551 × 10^−7^
*MWoutD			Solid187/refinement/36432				12,834	5685	0.545 × 10^−7^
*MWVS		Structural -> static structural (2)	Solid187/refinement/38304				13,478	5930	0.534 × 10^−7^

*MWD = microgripper with damping elements; *MWoutD = microgripper without damping elements; *MWVS = microgripper with V-shaped actuator.

**Table 10 micromachines-13-01460-t010:** Comparison of performance parameters of microgripper, obtained by simulation in ANSYS.

Actuator	Displacement (µm) at 2 V	Force (mN) at 2 V	∆T at 2 V	Natural Frequency (kHz)
MWoutD	1.830	70.151	111.53	14.899
MWD	2.426	73.61	111.52	37.994
MWVS	0.426	42.11	111.09	11.361

**Table 11 micromachines-13-01460-t011:** Equivalent von Mises stress values for MWD, and MWoutD.

Von Misses Stress (MPa)					
MWD					
A	B	C	D	E	F
24.2	34.8	68.65	18.0	37.93	57.26
MWoutD					
G	H	I	J	K	L
33.98	35.13	71.74	27.8	42.1	72.19

## Data Availability

Not applicable.

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
