# Peer review of "Z-Shaped Electrothermal Microgripper Based on Novel Asymmetric Actuator"

_micromachines, 2022, doi:10.3390/mi13091460_

Round 1

Reviewer 1 Report

The authors have proposed  a new design configuration to build a micro-gripper operated using asymmetric Z-shaped thermal actuator.  The idea is new and novel and can impact the field of particles manipulation processes. The manuscript fits the scope of the journal and can be of interest of  the readers. 

In the reviewer’s opinion, the novelty of the paper has bot been demonstrated in more detail and stated clearly. This part is important and should be emphasised. 

The manuscript structure needs a major revision because the way it written do not support the argument. The conclusion has not been added and I believe it would be beneficial to add a short section summaries the main findings of this manuscript and how to advance it in the further.

Indeed, the manuscript is within the scope of the journal and just need a cleanup to make it solid and comprehensive. The manuscript has several typos and needs revision. 

Author Response

Review of Actuators-438066

Z-shaped Electrothermal Microgripper based on Novel Asymmetric Actuator

Manuscript ID: micromachines-1873874

Respectable Reviewer 1:

We appreciate your valuable comments, due to them allow us to greatly improve the readability and content of the article, increasing notably its quality. The changes incorporated in response to your comments are highlighted to facilitate your review.

In the reviewer’s opinion, the novelty of the paper has both been demonstrated in more detail and stated clearly. This part is important and should be emphasized. 

Response: Thanks for your comment.

The title has been slightly modified to also reflect the novelty level of the microactuator.

In abstract and conclusion (recently added) sections it has been remarked the novelty level of the gripper geometry, which involves a Z-shaped beam and damping elements, as well as the new asymmetric microactuator pointing out the differences respected to the V-shaped actuator.

The manuscript structure needs a major revision because the way it written do not support the argument. The conclusion has not been added and I believe it would be beneficial to add a short section summary the main findings of this manuscript and how to advance it in the further.

Response: Thanks for your comment.

The applications derived from the performance parameters are given in lines 446-447, 451-454.

Regarding to future development of this work, in lines 455-459, the envisioned theoretical and experimental work is identified.

The Conclusions section has been added, as section 5, line 461-480.

Indeed, the manuscript is within the scope of the journal and just need a cleanup to make it solid and comprehensive. The manuscript has several typos and needs revision. 

                Response: Thanks for your comment.

The manuscript has been further revised by the authors.

Reviewer 2 Report

The authors present a paper titled “Z-shaped Electrothermal Microgripper based on Asymmetric Actuator”. There are a handful of specific technical concerns as below:

1) There are so many research articles previously reported on V an Z-shaped actuator where they discussed the trade off between force, displacement and applied voltage. The force  and displacement can be increased by changing the number of V  or Z shaped beams. What is the novelty of the current research work?

2) The asymmetrical Z-shaped actuator will generate non  linearity which is avoided in most of the application. For which application  the current research work is targeted for? 

Author Response

Review of Actuators-438066

Z-shaped Electrothermal Microgripper based on Novel Asymmetric Actuator

Manuscript ID: micromachines-1873874

Respectable Reviewer 2:

We appreciate your valuable comments, due to them allow us to greatly improve the readability and content of the article, increasing notably its quality. The changes incorporated in response to your comments are highlighted to facilitate your review.

1) There are so many research articles previously reported on V an Z-shaped actuator where they discussed the trade off between force, displacement and applied voltage. The force  and displacement can be increased by changing the number of V  or Z shaped beams. What is the novelty of the current research work?

Response: Thanks for your comment.

As you indicate, a great deal of research has been done, in particular on the V-shaped beam actuator. Our interest in designing the new actuator of asymmetric beams was to obtain a high force level, without the need to involve a considerable area, the design trade-off that is inherited, however, makes the displacement smaller. In summary, with the new actuator with simple geometry, we achieved a high force level, considering a very small area.

2) The asymmetrical Z-shaped actuator will generate non  linearity which is avoided in most of the application. For which application  the current research work is targeted for?

Response: Thanks for your comment.

The microgripper is designed to manipulate micro-objects, within a range of weight and geometric dimensions determined by the displacement and force generated in the microgripper jaws. Some immediate applications could be in research of materials or microdevices, for observation, or experimental test of stress, among others. The device has a relatively high gripping force with a compact design, making it suitable for potential applications as part of microinjection or drilling systems, along with other elements. However, for this possible application, future intensive research is required to plan strategies to cope with the generated nonlinearities in the microstructure.

These potential applications have been added in the manuscript in lines 446-447, 451-454.

Round 2

Reviewer 1 Report

The modified version strengthens the quality of the manuscript and I believe it deserves publishing in Micromachines in its present form. 

Reviewer 2 Report

I am satisfied with author's response.